# High frequency of potential phosphodiesterase type 5 inhibitor drug interactions in males with HIV infection and erectile dysfunction

**Jason M. Cota[1], Taylor M. Benavides[1], John D. Fields[1], Nathan Jansen[2], Anuradha Ganesan[3,4,5], Rhonda E. Colombo[3,5,6], Jason M. Blaylock[4], Ryan C. Maves[5,7], Brian K. Agan[3,5], Jason F. Okulicz[2] ***

1 University of the Incarnate Word, San Antonio, TX, United States of America, 2 San Antonio Military Medical Center, Fort Sam Houston, TX, United States of America, 3 Henry M. Jackson Foundation for the Advancement of Military Medicine, Bethesda, MD, United States of America, 4 Walter Reed National Military Medical Center, Bethesda, MD, United States of America, 5 Infectious Disease Clinical Research Program, Uniformed Services University of the Health Sciences, Bethesda, MD, United States of America, 6 Madigan Army Medical Center, Joint Base Lewis-McChord, WA, United States of America, 7 Naval Medical Center, San Diego, CA, United States of America

* jason.f.okulicz.mil@mail.mil

**Data Availability Statement:** Data for this study are available from the Infectious Disease Clinical Research Program (IDCRP), headquartered at the

## Abstract

### Objectives

We sought to determine the prevalence of phosphodiesterase type 5 inhibitor (PDE-5) mediated drug-drug interactions (DDIs) in males with HIV infection receiving antiretroviral therapy (ART) and identify factors associated with PDE-5-mediated DDIs.

### Methods

Male US Military HIV Natural History Study participants diagnosed with erectile dysfunction (ED) and having a PDE-5 inhibitor and potentially-interacting ART co-dispensed within 30 days were included. DDIs were defined according to criteria found in published guidelines and drug information resources. The primary outcome of interest was overall PDE-5 inhibitor-mediated DDI prevalence and episode duration. A secondary logistic regression analysis was performed on those with and without DDIs to identify factors associated with initial DDI episode.

### Results

A total of 235 male participants with ED met inclusion criteria. The majority were White (50.6%) or African American (40.4%). Median age at medication co-dispensing (45 years), duration of HIV infection (14 years), and duration of ED (1 year) did not differ between the two groups (p>0.05 for all). PDE-5 inhibitors included sildenafil (n = 124), vardenafil (n = 99), and tadalafil (n = 14). ART regimens included RTV-boosted protease inhibitors (PIs) atazanavir (n = 83) or darunavir (n = 34), and COBI-boosted elvitegravir (n = 43). Potential DDIs

Uniformed Services University of the Health Sciences (USU), Department of Preventive Medicine and Biostatistics. The Informed Consent Document under which the HIV Natural History Study data were collected specifies that each use of the data will be reviewed by the Institutional Review Board. Furthermore, the data set may include Military Health System data collected under a Data Assurance Agreement that requires accounting for uses of the data. Data requests may be sent to: Address: 6270A Rockledge Drive, Suite 250, Bethesda, MD 20817. Email: contactus@idcrp.org.

**Funding:** Support for this work (IDCRP-000-03) was provided by the Infectious Disease Clinical Research Program (IDCRP), a Department of Defense (DoD) program executed through the Uniformed Services University of the Health Sciences. This project has been funded in whole, or in part, with federal funds from the National Institute of Allergy and Infectious Diseases, National Institutes of Health (NIH), under Inter-Agency Agreement Y1-AI-5072. The funders had no role in study design, data collection and analysis, decision to publish, or preparation of the manuscript.

**Competing interests:** The authors have declared that no competing interests exist.

occurred in 181 (77.0%) participants, of whom 122 (67.4%) had multiple DDI episodes. The median DDI duration was 8 (IQR 1–12) months. In multivariate analyses, non-statistically significant higher odds of DDIs were observed with RTV-boosted PIs or PI-based ART (OR 2.13, 95% CI 0.85–5.37) and in those with a diagnosis of major depressive disorder (OR 1.74, 95% CI 0.83–3.64).

## Conclusions

PDE-5-mediated DDIs were observed in the majority of males with HIV infection on RTV- or COBI-boosted ART in our cohort. This study highlights the importance of assessing for DDIs among individuals on ART, especially those on boosted regimens.

## Introduction

Men with human immunodeficiency virus (HIV) infection have a prevalence of erectile dysfunction (ED) that may be as high as 50% [1–3]. Phosphodiesterase type 5 (PDE-5) inhibitors such as sildenafil, tadalafil, and vardenafil are the drugs of choice for ED. Given that each of these PDE-5 inhibitors is primarily metabolized by the cytochrome P450 3A4 isoenzyme (CYP3A4), drug information databases (hiv-druginteractions.org) and antiretroviral therapy (ART) guidelines warn against co-administration with strong CYP3A4 inhibitors [4–7]. Protease inhibitors (PIs) and pharmacokinetic boosters such as ritonavir (RTV) and cobicistat (COBI) strongly inhibit CYP3A4 drug metabolism [6, 8, 9]. Therefore, resources recommend PDE-5 inhibitor dose reduction when initiating therapy with these potentially-interacting ART regimens or recommend lower starting PDE-5 inhibitor doses in patients already receiving this ART to avoid excessive PDE-5 inhibitor exposure. These recommendations are not only based on the theoretical mechanism of this CYP3A4-mediated interaction, but also on pharmacokinetic studies confirming substantially-elevated PDE-5 inhibitor levels with co-administration of PIs, RTV, or COBI. A four-fold increase in sildenafil systemic exposure results when given with PIs alone or RTV-boosted PIs. A greater than two-fold increase in sildenafil maximum serum concentrations was observed with COBI-boosted ART co-administration [10–12]. Excessive PDE-5 inhibitor exposure may result in clinically significant adverse effects such as hypotension, syncope, visual disturbances and/or priapism. PDE-5 inhibitor DDIs may be especially concerning because conflicting data suggest an increase in ED risk with prolonged HIV and PI use [3, 13, 14]. While several studies have documented the prevalence of clinically-significant DDIs in HIV-infected patients, none have specifically focused on potential interactions between PDE-5 inhibitors and ART [15–20]. We evaluated the prevalence and factors associated with PDE-5-mediated DDIs in men with ED and HIV infection on potentially-interacting ART.

## Materials and methods

### Study design

The US Military HIV Natural History (NHS) is a prospective observational cohort of active duty US military members and beneficiaries with HIV infection. Participants are evaluated approximately every 6–12 months with data collected for demographics, laboratory studies, medical diagnoses, and medications. All participants were ≥18 years of age and provided informed written consent for this IRB-approved study.

The NHS database was queried to identify male participants with a diagnosis of ED recorded between 2001 and 2016. Those with an ED diagnosis receiving a prescription for a PDE-5 inhibitor (sildenafil, vardenafil, or tadalafil) and a potentially-interacting ART regimen co-dispensed within 30 days were included. Potentially-interacting ART regimens included those that contained a PI, RTV, or COBI. Patient characteristics including age, race/ethnicity, initial HIV diagnosis date, and initial ED diagnosis date were collected. Race and ethnicity data were collected because previous studies have identified differences in the risk factors for and prevalence of severe erectile dysfunction [21]. While all patients were diagnosed with ED, collected comorbidities only included those previously identified as posing a high risk for severe ED such as cardiovascular disease, mental health disorders, diabetes, and smoking history. Comprehensive prescription information available for analysis included prescription fill dates, medication name, medication dose, and quantity dispensed. Additional medications known to affect CYP3A4 metabolism were captured if co-prescribed within 30-days of PDE-5 inhibitor dispensing date. These included 3A4 inhibitors (macrolides, azole antifungals, and statins) or those with 3A4 induction potential (rifamycins and anti-epileptic drugs).

## Outcomes

The primary analysis sought to describe the prevalence and duration of DDIs between PDE-5 inhibitors and ART regimens containing a strong CYP3A4 inhibitor (PI, RTV, or COBI). A DDI was identified when a PDE-5 inhibitor was co-dispensed within 30 days of potentially-interacting ART using the following consistent definitions found across published guidelines and drug interaction programs [4–7]. For a patient with a past and current potentially-interacting ART prescription history who received a new first prescription for a PDE-5 inhibitor, a DDI was documented if: 1) the initial sildenafil dose exceeded 25 mg, 2) the initial vardenafil or tadalafil dose exceeded 2.5 mg, or 3) the tadalafil dose exceeded 10 mg at any time during the co-administration period. For those with a past and current PDE-5 inhibitor prescription history who received a new first prescription for a potentially-interacting ART regimen, a DDI was documented if the PDE-5 inhibitor dose was not discontinued or the dose was not reduced within 30 days of the new prescription for any PI-based ART (RTV-boosted or un-boosted) or COBI-boosted ART. The duration of a single discrete DDI episode was calculated by using the days supply listed on the prescription record. An additional DDI episode was captured if the aforementioned DDI definitions were met following a gap in prescription overlap greater than 90 days following the documented days supply.

## Statistical analysis

Descriptive statistics were used to determine DDI prevalence and episode duration. In order to determine what factors were associated with the presence of a DDI, patients were divided into a DDI and a non-DDI cohort for logistic regression analysis. The following factors for a first DDI occurrence were evaluated using univariate logistic regression: age, race/ethnicity, time from HIV diagnosis to first DDI, time from ED diagnosis to first DDI, comorbidities that increase risk for more severe ED, first PDE-5 inhibitor prescription after RTV- or COBI-boosted ART, time from HIV diagnosis to first DDI, and time from ED diagnosis to first DDI. Factors with a p-value less than 0.2 in univariate analysis were included in a multivariate logistic regression model to identify associations associated with first DDI occurrence. Chi-square test was used to analyze categorical data. Continuous variables were assessed for normality using Shapiro-Wilk W test and Wilcoxon rank sum was used to compare the non-normally distributed data. A p-value less than 0.05 was considered statistically significant. Statistics were calculated using JMP® software (JMP Pro®, Version 14.1, SAS Institute Inc., Cary, NC, 1989–2019).

## Results

Over the 16-year study period, 499 male NHS participants with an ED diagnosis were identified with a total of 137,632 prescriptions records available (Fig 1). Of the 35,918 ART prescriptions dispensed, 21.2% were for potentially-interacting ART (Fig 2). There were 7,590 PDE-5 inhibitor prescriptions dispensed with 52% for sildenafil, 40% for vardenafil, and 8% for tadalafil. Overall, 235 participants met criteria for study inclusion (Fig 1). The majority of participants were White (50.6%) with a median age of 45 (IQR 40–51) years at first co-dispensed PDE-5 inhibitor and ART. Participant characteristics were similar in the DDI and non-DDI groups (p>0.05 for all).

### Outcomes

A total of 181 (77.0%) participants met the DDI definition (Table 1), of whom 122 (51.9%) had multiple DDI episodes. The median DDI duration was 8 (IQR 1–12) months. The majority of DDIs occurred in individuals already on PDE-5 inhibitors followed by the addition of ART (55.8%). Among initial DDI episodes, sildenafil doses were 100 mg (55.4%) and 50 mg (44.6%) whereas vardenafil doses were 10 mg (37.3%), 20 mg (32%), and 5 mg (30.7%). All 14 tadalafil prescriptions met criteria for a DDI when co-administered with potentially-interacting ART including 7, 6, and 1 prescriptions dispensed for 10 mg, 20 mg, and 5 mg doses, respectively. In two patients in the DDI cohort, simvastatin prescriptions were dispensed along with sildenafil and RTV-boosted regimens. As a CYP3A4 inhibitor, simvastatin would be expected to further exacerbate the PDE-5 inhibitor-RTV interaction. In both instances, the simvastatin prescription was discontinued and replaced by an alternative statin without CYP3A4 interaction potential. In one of these patients, one additional 90-day supply sildenafil prescriptions were dispensed. In the second patient, sildenafil was switched to vardenafil at an inappropriate dose of 10 mg. No patients in the non-DDI group received additional prescriptions known to affect CYP3A4 metabolism within 30 days of the co-dispensing of a PDE-5 inhibitor and potentially-interacting ART.

Most DDIs occurred during the 2005–2008 period and involved RTV-boosted regimens (Fig 3). From 2012–2016, 72.7% of co-administrations of PDE-5 inhibitors and potentially-

**Fig 1.** Flow chart of participants screened for eligibility and included in study.

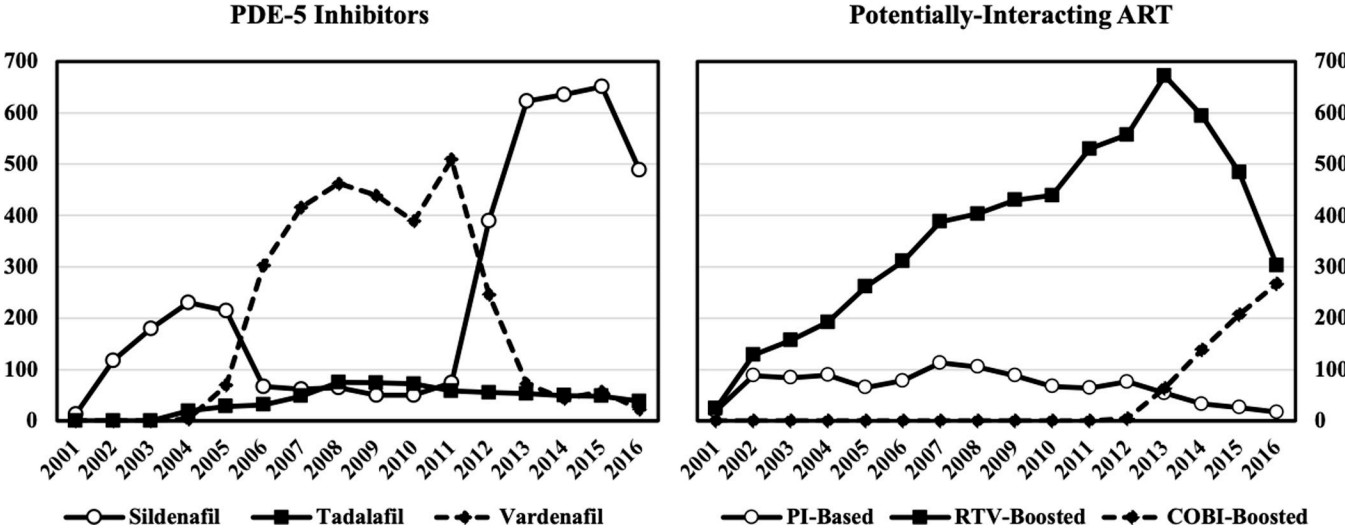

**Fig 2. Number of PDE-5 inhibitor and potentially-interacting ART dispensed by year of study.**

interacting ART met criteria for a DDI and increasingly involved the COBI-boosted elvitegravir regimen. PDE-5 inhibitor use peaked in 2015 with sildenafil representing 85% of PDE-5 inhibitors dispensed. RTV-boosted ART regimens peaked in 2013; the subsequent decline in RTV-boosted regimens was offset with an increase in COBI-boosted elvitegravir, which peaked during the last study year in 2016.

Among participant factors in univariate analyses that met criteria for inclusion in the multivariate model, age less than 45 and major depressive disorder were associated with increased odds of DDIs; whereas dyslipidemia was associated with decreased DDI risk (Table 2). There were four medication factors that were included in the multivariate analysis. Major depressive disorder and PI-based ART were associated with an increased DDI risk, but these factors did not reach statistical significance in multivariate analyses.

## Discussion

A clinically-significant DDI involving ART may occur in approximately 1 in 3 to 4 persons living with HIV [7, 15, 17, 18]. This is the first study specifically addressing the prevalence of PDE-5 inhibitor-mediated DDIs in men with HIV infection. Studies confirming increased PDE-5 inhibitor concentrations with coadministration of PIs and pharmacokinetic boosters have been well documented since these ED treatment agents were first available for clinical use [12, 22–24]. At least two published case reports describe deaths associated with DDIs involving PDE-5 inhibitors and RTV-boosted ART [23, 25]. Such consequences make the recommendations for PDE-5 inhibitor dose reduction and slow up-titration essential when these ED agents are co-prescribed with strong CYP 3A4 inhibitors such as PIs, RTV, and COBI.

DDI management recommendations may be found in clinical practice guidelines and with use of dedicated online antiretroviral DDI checkers (hiv-druginteractions.org) [6, 7]. Despite this longstanding knowledge, PDE-5 inhibitor doses exceeded recommendations in more than 75% of HIV-infected men with ED receiving potentially-interacting ART in this study. In addition, approximately 60% of these patients received a PDE-5 inhibitor dose that was more than 4 to 8 times higher than the recommended starting dose. This would yield initial systemic exposures greater than 10 times higher than maximum recommended PDE-5 inhibitor doses when given alone.

**Table 1. Participant characteristics.**

| | All (n = 235) | DDI Group (n = 181) | Non-DDI Group (n = 54) | p-value |
|---|---|---|---|---|
| Median age at first DDI (years) | 45 (40–51) | 45 (40–51) | 46 (41–51) | 0.75 |
| Race/ethnicity | | | | |
| White | 119 (50.6) | 92 (50.8) | 27 (50.0) | 0.91 |
| African American | 95 (40.4) | 72 (39.8) | 23 (42.6) | 0.71 |
| Hispanic/Puerto Rican/Mexican | 11 (4.7) | 10 (5.5) | 1 (1.9) | 0.26 |
| High risk co-morbidities | | | | |
| Dyslipidemia | 138 (58.7) | 101 (55.8) | 37 (68.5) | 0.10 |
| Hypertension | 72 (30.6) | 56 (30.9) | 16 (29.6) | 0.85 |
| Major depressive disorder | 68 (28.9) | 57 (31.5) | 11 (20.4) | 0.11 |
| Generalized anxiety disorder | 29 (12.3) | 23 (12.7) | 6 (11.1) | 0.94 |
| Type 2 diabetes mellitus | 19 (8.1) | 16 (8.8) | 3 (5.6) | 0.62 |
| Tobacco use | 30 (12.8) | 23 (12.7) | 7 (13.0) | 0.85 |
| Two or more high risk co-morbidities | 45 (19.1) | 34 (18.8) | 11 (20.4) | 0.79 |
| Median time from HIV diagnosis to first co-administration (years) | 14 (8–19) | 15 (9–18) | 13 (6–21) | 0.75 |
| Median time from ED diagnosis to first co-administration (days) | 379 (77–1622) | 381 (89–1601) | 360 (54–1694) | 0.53 |
| Median number of prescriptions at time of first co-administration | 6 (4–15) | 6 (4–15) | 5 (4–15) | 0.75 |
| PDE-5 inhibitor started before potentially-interacting ART | 134 (57.0) | 101 (55.8) | 33 (61.1) | 0.49 |
| **PDE-5 inhibitor** | | | | |
| sildenafil | 124 (52.8) | 92 (50.8) | 32 (59.3) | 0.28 |
| vardenafil | 99 (42.1) | 77 (42.5) | 22 (40.7) | 0.81 |
| tadalafil | 14 (6.0) | 14 (7.7) | 0 (0) | 0.12 |
| **Type of potentially-interacting ART** | | | | |
| RTV-boosted regimens | 162 (68.9) | 125 (69.1) | 37 (68.5) | 0.94 |
| atazanavir | 83 (35.3) | 68 (37.6) | 15 (27.8) | 0.19 |
| darunavir | 34 (14.5) | 23 (12.7) | 11 (20.4) | 0.16 |
| lopinavir | 35 (14.9) | 29 (16.0) | 6 (11.1) | 0.37 |
| Cobicistat-boosted elvitegravir | 43 (18.3) | 29 (16.0) | 14 (25.9) | 0.10 |

Data expressed and number (%) or median (interquartile range)

DDI, drug-drug interaction; ED, erectile dysfunction; PDE-5, phosphodiesterase type 5; ART, antiretroviral therapy; RTV, ritonavir

It is unlikely that incomplete medical records contributed to the high frequency of DDIs in this cohort. Lack of clinician and patient awareness of potential interactions combined with underreporting of adverse consequences with this combination may be reasons why we observed such a high frequency of PDE-5 inhibitor-mediated DDIs. This is consistent with one study in which physicians were unable to correctly identify two-thirds of clinically-significant ART-mediated DDIs [26]. Similar findings from a separate study showed that DDIs were more likely to occur and be mismanaged when the interacting drug was prescribed by a provider not primarily involved in HIV care and ART prescribing [27]. Sildenafil is also indicated for pulmonary hypertension; co-administration of sildenafil and ritonavir is contraindicated in this setting. However, after review of the prescribed dosing regimens is it is unlikely that NHS participants were receiving sildenafil for pulmonary hypertension. Vardenafil prescriptions increased from 2006 to 2011; however, the percent

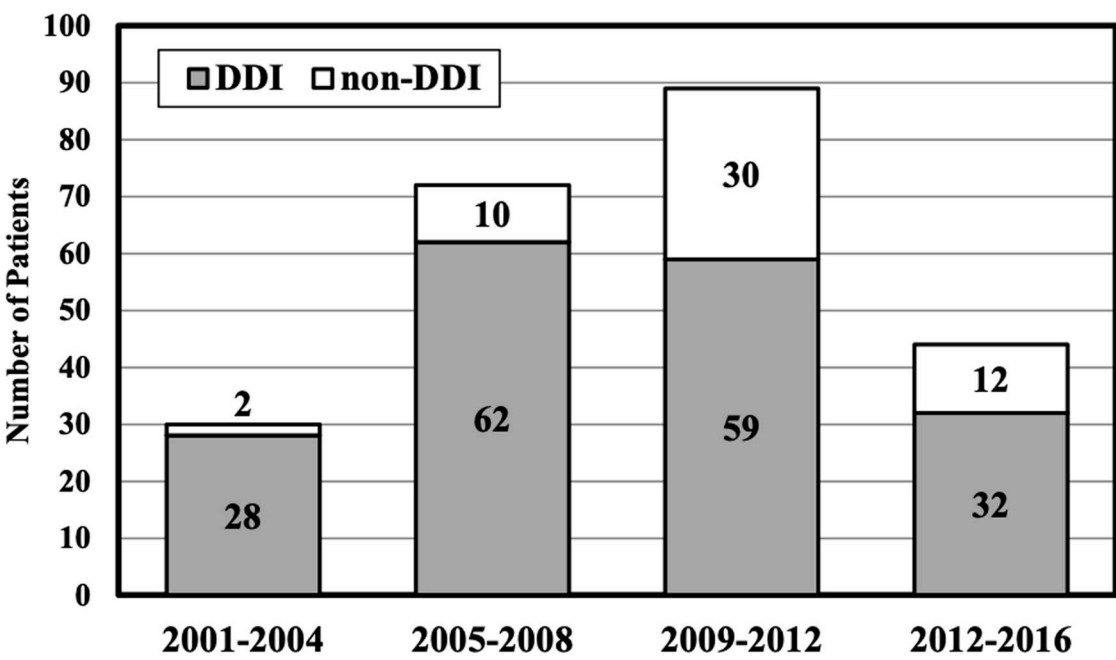

**Fig 3. Potentially-interacting co-administrations per 4-year study period.**

of DDIs did not increase during this period when vardenafil was used along with potentially-interacting ART. PDE-5 inhibitor sharing and "recreational use" has been previously reported, but was not assessed in this study [28, 29].

Major depressive disorder and receipt of any PI-based ART were associated with an elevated, but non-statistically-significant increased risk of a PDE-5 inhibitor-mediated DDIs. Other studies have shown an increased risk of DDIs with use of PI-based or RTV-boosted ART regimens [15–18]. Patients were included in logistic regression models based on the first date of PDE-5 inhibitor and potentially-interacting ART co-administration. Using this approach and given the 2001 start date of this longitudinal database, a higher number of patients incorporated into the models were receiving any PI-based ART compared to newer COBI-boosted regimens, which were unavailable before 2012. However, more than 50% of patients included in our model had additional discrete DDIs with different PDE-5 inhibitors and/or newer ART regimens occur later in the study period. It is unclear how a different approach for model inclusion would have affected the detection of independent risk factors for DDIs.

## Strengths

The strengths of this study include its large cohort size, large number of screened individuals, and the comprehensive prescription database maintained for participants enrolled in the US Military HIV Natural History Study.

## Limitations

This study was limited by the retrospective design. Unfortunately, access to specific documented instances of hypotension, syncope, visual disturbances and/or priapism that may be associated with PDE-5 inhibitor-mediated DDIs were not available.

**Table 2. Factors associated with PDE-5 inhibitor-mediated drug-drug interactions.**

| Univariate Logistic Regression | | OR | 95% CI | p-value |
|---|---|---|---|---|
| Participant Factors | | | | |
| | Age <45 years at co-dispensing | 1.55 | 0.83–2.89 | 0.16^ |
| | HIV >14 years at co-dispensing | 1.33 | 0.72–2.44 | 0.36 |
| | ED >1 year at co-dispensing | 1.05 | 0.58–1.94 | 0.86 |
| | Dyslipidemia | 0.58 | 0.30–1.10 | 0.10^ |
| | Hypertension | 1.06 | 0.55–2.07 | 0.85 |
| | Type 2 diabetes mellitus | 1.65 | 0.46–5.88 | 0.44 |
| | Generalized anxiety disorder | 2.00 | 0.67–6.03 | 0.22 |
| | Major depressive disorder | 1.80 | 0.86–3.74 | 0.12^ |
| | Smoking history | 0.75 | 0.24–1.55 | 0.25 |
| | >2 high-risk co-morbidities | 0.90 | 0.42–1.93 | 0.80 |
| | African American | 0.89 | 0.48–1.65 | 0.71 |
| Medication Factors | | | | |
| | Receipt of ART before PDE-5 inhibitor | 1.25 | 0.67–2.32 | 0.49 |
| | Polypharmacy (>5 medications) | 1.41 | 0.75–2.65 | 0.29 |
| | Any PI-based ART | 1.83 | 0.38–3.79 | 0.10^ |
| | RTV-boosted atazanavir | 1.56 | 0.80–3.05 | 0.19^ |
| | RTV-boosted darunavir | 0.57 | 0.26–1.26 | 0.16^ |
| | RTV-boosted lopinavir | 1.53 | 0.60–3.90 | 0.38 |
| | Sildenafil | 0.71 | 0.38–1.32 | 0.18^ |
| | Vardenafil | 1.08 | 0.58–2.00 | 0.81 |
| **Multivariate Logistic Regression** | | **OR** | **95% CI** | **p-value** |
| RTV-boosted atazanavir | | 1.07 | 0.47–2.44 | 0.87 |
| Dyslipidemia | | 0.88 | 0.45–1.72 | 0.72 |
| Sildenafil | | 0.87 | 0.46–1.72 | 0.69 |
| Age<45 | | 1.27 | 0.66–2.43 | 0.47 |
| Major depressive disorder | | 1.74 | 0.83–3.64 | 0.14 |
| RTV-boosted darunavir | | 0.48 | 0.19–1.23 | 0.13 |
| RTV-boosted PI or PI-based ART | | 2.13 | 0.85–5.37 | 0.11 |

ART, antiretroviral therapy; PDE-5, phosphodiesterase type 5; RTV, ritonavir

^Univariate factors with a P<0.2 were included in the multivariate analysis

## Conclusion

Drug interactions related to ED and ART therapy were identified in almost half of men with HIV infection receiving PDE-5 inhibitors for ED in our cohort. The high prevalence of co-administered ART and PDE-5 inhibitors with known interactions in this cohort of men with excellent access to healthcare highlights the need for ongoing multidisciplinary education on the importance of assessing for interactions when initiating or modifying medications in people with HIV on ART.

## Acknowledgments

**Disclaimer:** The content of this publication is the sole responsibility of the authors and does not necessarily reflect the views or policies of the Uniformed Services University of the Health Sciences, Brooke Army Medical Center, the US Army Medical Department, the US Army Office of the Surgeon General, the Department of the Navy, the Department of the Army, the

Department of the Air Force, the Department of Defense, or the US Government. Mention of trade name, commercial products, or organizations does not imply endorsement by the US Government.

The authors are employees of the U.S. Government. This work was prepared as part of their official duties. Title 17 U.S.C. 105 provides that 'Copyright protection under this title is not available for any work of the United States Government.' Title 17 U.S.C. 101 defines a United States Government work as a work prepared by a military service member or employee of the United States Government as part of that person's official duties.

## Author Contributions

**Conceptualization:** Jason M. Cota, Nathan Jansen.

**Data curation:** Taylor M. Benavides, John D. Fields, Nathan Jansen, Jason F. Okulicz.

**Formal analysis:** Jason M. Cota, Taylor M. Benavides, John D. Fields, Jason F. Okulicz.

**Funding acquisition:** Brian K. Agan.

**Investigation:** Jason M. Cota, Nathan Jansen, Rhonda E. Colombo, Jason M. Blaylock, Ryan C. Maves, Brian K. Agan, Jason F. Okulicz.

**Methodology:** Jason M. Cota, John D. Fields, Nathan Jansen, Anuradha Ganesan, Brian K. Agan, Jason F. Okulicz.

**Project administration:** Nathan Jansen, Brian K. Agan, Jason F. Okulicz.

**Resources:** Anuradha Ganesan, Rhonda E. Colombo, Ryan C. Maves, Brian K. Agan, Jason F. Okulicz.

**Supervision:** Jason M. Cota, Jason F. Okulicz.

**Validation:** Nathan Jansen, Anuradha Ganesan, Jason M. Blaylock, Ryan C. Maves, Brian K. Agan.

**Visualization:** Jason F. Okulicz.

**Writing – original draft:** Jason M. Cota, Jason F. Okulicz.

**Writing – review & editing:** Jason M. Cota, Taylor M. Benavides, John D. Fields, Nathan Jansen, Anuradha Ganesan, Rhonda E. Colombo, Jason M. Blaylock, Ryan C. Maves, Brian K. Agan, Jason F. Okulicz.

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
