## [Decision Letter · Decision Letter 0]

11 Jan 2021

PONE-D-20-32474

High Frequency of Potential Phosphodiesterase Type 5 Inhibitor Drug Interactions in Males with HIV Infection and Erectile Dysfunction

PLOS ONE

Dear Dr. Okulicz,

Thank you for submitting your manuscript to PLOS ONE. After careful consideration, we feel that it has merit but does not fully meet PLOS ONE’s publication criteria as it currently stands. Therefore, we invite you to submit a revised version of the manuscript that addresses the points raised during the review process.

We look forward to receiving your revised manuscript.

Kind regards,

Professor Kwasi Torpey, MD PhD MPH

Academic Editor

PLOS ONE

Additional Editor Comments:

The manuscript seeks to determine the prevalence of PDE-5 inhibitor drug drug interactions with antiretroviral therapy among men with erectile dysfunction. It is a retrospective cohort and applying a case control approach. There are major methodological concerns that needs to be addressed

1. Cases and controls were selected from the cohort. Cases were those who had DDIs as defined. The controls were those without the DDIs. Given that several factors may influence DDIs it is important for the authors to elaborately describe the cases and whether the controls were matched or unmatched?. Were there other co-morbid conditions and medications apart from ART? Were these matched? Case control design is used when a condition is rare/uncommon, however in this manuscript, the cases were actually more than the controls in >3:1.

2. Patient characteristics Page 5 Line 75 mentions co-morbidities however no information on the co-morbid conditions and the associated medications is provided in the results section. PDE 5 inhibitors is known to have interactions with not only ART but antibiotics like erythromycin, clarithromycin, antifungals, hypertensive medications, alcohol etc. There is no evidence presented on these associated factors in the manuscript.

3. Definition of DDI: The authors define a DDI was documented if: 1) the initial sildenafil dose exceeded 25 mg, 2) the initial vardenafil or tadalafil dose exceeded 2.5 mg, or 3) the tadalafil dose exceeded 10 mg at any time during the co-administration period. For those receiving a PDE-5 inhibitor who received a new first prescription for a potentially-interacting ART regimen, a DDI was identified if the PDE-5 inhibitor dose was not discontinued or the dose was not reduced within 30 days of the new prescription for any PI-based ART (RTV-boosted or un-boosted) or COBI-boosted ART. The scientific basis of this criteria is unclear. For example Sildenafil is given at a dose from 25mg to100mg to get the desirable effect. This may be be due to specific patient related factors or idiosyncratic. It is therefore problematic to assume that a dose of more than 25mg may be a potential case of DDI. The criteria of the dose not discontinued or reduced should be adequately explained

4. Additional DDI captured for gaps in medication should adequately elaborated Page 5 Line 89

Others

5. Description of the metabolic pathway/mechanism of action: The introduction must include why RTV or COBI is likely to increase the levels of PDE 5 inhibitors. Is it through the cytochrome P450 enzyme complex or another mechanism? This should be described.

6. Review of point 1 to 4 may influence the outcomes./results and subsequently the conclusion

Journal Requirements:

2. Please provide a justification as to why participant ethnicity was studied as a variable. Furthermore, please note that according to our submission guidelines (http://journals.plos.org/plosone/s/submission-guidelines), outmoded terms and potentially stigmatizing labels should be changed to more current, acceptable terminology. For example: “Caucasian” should be changed to “white” or “of [Western] European descent” (as appropriate).

Finally please provide a citation to the guidelines which were used to identify DDIs [line 81].

Reviewers' comments:

Reviewer's Responses to Questions

**Comments to the Author**

1. Is the manuscript technically sound, and do the data support the conclusions?

Reviewer #1: No

2. Has the statistical analysis been performed appropriately and rigorously? 

Reviewer #1: No

3. Have the authors made all data underlying the findings in their manuscript fully available?

Reviewer #1: Yes

4. Is the manuscript presented in an intelligible fashion and written in standard English?

Reviewer #1: Yes

5. Review Comments to the Author

Reviewer #1: This study reported on drug-drug interaction (DDI) between phosphodiesterase type 5 (PDE-5) inhibitor and antiretroviral therapy (ART) in males with HIV infection and erectile dysfunction (ED) diagnosis. Generally, in this retrospective nature of the present study, causality cannot be inferred. They selected patients receiving both treatments among patients with HIV infection and ED, and defined those with high doses of PDE-5 inhibitor by a DDI group. Such patients may require its high doses due to disease severity or patient potential confounders, and I wonder why they have a conclusion that these patients met ED due to DDIs or they assessed other DDIs-related adverse effects such as hypotension and syncope. Additionally, how many patients taking both PDE-5 inhibitor and ART did not meet ED during the same period? Among these patients without ED, how many patients were considered as DDIs group, namely high doses of PDE-5 inhibitor? More detailed description regarding interaction between these treatments and adverse effects should be addressed. Otherwise, authors should downplay the interpretation of their results. I have following major concerns:

1. Patients needed higher doses of these disease modifying treatments might be likely to have sicker HIV conditions; hence this drug-drug interaction could be indispensable for managing such HIV patients. However, their description table lacked patient factors.

2. In multiple logistic regression model, all retained variables were medication factors (RTV-boosted atazanavir, Sildenafil, RTV-boosted, darunavir, RTV-boosted PI or PI-based ART). I think that these variables may be strongly correlated since patients with more disease severity require higher doses of these HIV-related treatments. Did you look at the multicollinearity of such medication variables?

6. PLOS authors have the option to publish the peer review history of their article (what does this mean?). If published, this will include your full peer review and any attached files.

Reviewer #1: **Yes: **Masatake Kobayashi

---

## [Author Response · Author response to Decision Letter 0]

5 Apr 2021

Please see response to reviewers document.

---

## [Editor Report · Decision Letter 1]

12 Apr 2021

High Frequency of Potential Phosphodiesterase Type 5 Inhibitor Drug Interactions in Males with HIV Infection and Erectile Dysfunction

PONE-D-20-32474R1

Dear Dr. Okulicz,

We’re pleased to inform you that your manuscript has been judged scientifically suitable for publication and will be formally accepted for publication once it meets all outstanding technical requirements.

Kind regards,

Professor Kwasi Torpey, MD PhD MPH

Academic Editor

PLOS ONE

Additional Editor Comments (optional):

Manuscript significant improved and comments satisfactorily addressed
---

## [Editor Report · Acceptance letter]

19 Apr 2021

PONE-D-20-32474R1 

High Frequency of Potential Phosphodiesterase Type 5 Inhibitor Drug Interactions in Males with HIV Infection and Erectile Dysfunction 

Dear Dr. Okulicz:

I'm pleased to inform you that your manuscript has been deemed suitable for publication in PLOS ONE. Congratulations! Your manuscript is now with our production department. 

Kind regards, 

on behalf of

Professor Kwasi Torpey 

Academic Editor

PLOS ONE